# “I Think Deprescribing as a Whole Is a Gap!”: A Qualitative Study of Student Pharmacist Perceptions about Deprescribing

**DOI:** 10.3390/geriatrics7030060

**Published:** 2022-06-02

**Authors:** Sydney P. Springer, Alina Cernasev, Rachel E. Barenie, David R. Axon, Devin Scott

**Affiliations:** 1Department of Pharmacy Practice, University of New England School of Pharmacy, Westbrook College of Health Professions, 716 Stevens Ave, Portland, ME 04013, USA; sspringer1@une.edu; 2Department of Clinical Pharmacy and Translational Science, University of Tennessee Health Science Center, College of Pharmacy, 301 S. Perimeter Park Dr., Suite 220, Nashville, TN 37211, USA; 3University of Tennessee Health Science Center, College of Pharmacy, Memphis, TN 38163, USA; rbarenie@uthsc.edu; 4Department of Pharmacy Practice & Science, College of Pharmacy, University of Arizona, 1295 N Martin Ave, Tucson, AZ 85721, USA; axon@pharmacy.arizona.edu; 5Teaching and Learning Center, Department of Academic, Faculty and Student Affairs, University of Tennessee Health Science Center, 920 Madison, Suite 424, Memphis, TN 38163, USA; dscott50@uthsc.edu

**Keywords:** deprescribing, student pharmacist, United States of America, pharmacy education

## Abstract

Introduction: Older adults often manage multiple medications simultaneously, contributing to significant pill burden. Pill burden is a major concern for both patients and providers alike, and student pharmacists may play a role in decreasing that burden. Few studies exist evaluating student pharmacists’ roles in and perceptions of deprescribing in the healthcare team. Thus, the aim of this study was to explore student pharmacist perceptions regarding deprescribing in their pharmacy curricula. Methods: This study used a focus group discussion (FGD) methodology to facilitate discussion on deprescribing among student pharmacists. The theory of planned behavior (TPB) informed the conceptualization of this study, data collection, and thematic analysis. Student pharmacists enrolled in three different colleges of pharmacy across various geographical regions of the U.S. were recruited to participate in the study. Data collection occurred in the Fall of 2021, and recruitment proceeded until thematic saturation was achieved. The audio recordings were transcribed verbatim, and the transcripts were uploaded into Dedoose^®^, a qualitative software that facilitated the data analysis. The inductive codes were grouped into categories based on similarities that resulted in the themes. Results: Three colleges, totaling 1366 student pharmacists across different geographic regions of the U.S., were invited to participate in this study (UTHSC, N = 682; UNE, N = 158, University of Arizona, N = 526). Twenty-six student pharmacists participated in four FGDs. Of 26 participants, fourteen self-identified as male and two declined to state their gender identity. The mean age was 24 years old, with participants ranging from 21 to 37 years old. Thematic analysis revealed two major themes: (1) student pharmacists indicated that they possessed limited information about the deprescribing process, which is best illustrated by the following quote: “I think deprescribing as a whole is a gap!”; and (2) student pharmacists recommended increasing emphasis on deprescribing in pharmacy curricula. Conclusions: Student pharmacists identified few educational experiences on deprescribing in their curriculum while demonstrating a strong desire for more emphasis on deprescribing in the pharmacy curricula. This study highlights an opportunity to improve the integration of deprescribing education into pharmacy curricula, and colleges of pharmacy should evaluate whether, where, and to what extent the incorporation of this topic into their curricula is appropriate.

## 1. Introduction

Older adults often manage multiple medications simultaneously, with over one-third managing more than five medications, contributing to significant pill burden [1,2]. Polypharmacy, defined commonly as being on more medications than clinically indicated, is common in older adults. As we age, we are more likely to develop, and subsequently manage, multiple chronic conditions, such as high blood pressure, high cholesterol, and diabetes [2]. The medications used to manage these multiple conditions are often considered potentially inappropriate per the American Geriatric Society (AGS) Beers Criteria [3]. Inappropriate medications are associated with poor medical outcomes including increases in emergency department visits, hospitalizations, and morbidity [4,5].

One opportunity to mitigate the harms associated with inappropriate medication use is deprescribing. Deprescribing is defined as “the systematic process of identifying and discontinuing drugs in instances in which existing or potential harms outweigh existing or potential benefits within the context of an individual patient’s care goals, current level of functioning, life expectancy, values, and preferences,” and it can provide an opportunity to optimize a patient’s medication use [5]. Deprescribing has been shown to reduce potentially inappropriate or unnecessary medications without worsening symptoms [6]. In a small study of 65 older adults with type 2 diabetes, the simplification and reduction of insulin regimens was associated with a reduction in hypoglycemia, an improvement in the Diabetes-related distress score, and no worsening of glycemic control [7]. A pragmatic randomized trial assessing the impact of statin discontinuation in adults near the end of life found that the discontinuation of statin therapy was associated with improved quality of life and had no excess risk of 60-day mortality amongst those with primary and secondary cardiovascular disease prevention indications [8]. Currently, however, no standard of training and instruction on deprescribing for student pharmacists exists in the U.S [9].

Additionally, there are few studies conducted with student pharmacists to explore the ways in which deprescribing and the role of pharmacists in deprescribing are addressed in the pharmacy curricula. In one study, Woodall et al. conducted a cross-sectional survey of geriatric pharmacy education in the U.S [10]. They found that only 35% of institutions had a required geriatric course, despite 58% of institutions indicating that they were “somewhat confident” or “very confident” that their institution was preparing their students to administer quality care for older adults [10]. Another survey-based study assessed deprescribing in the curricula of pharmacy, nursing, and medical programs [11]. The study reported that student pharmacists perceived higher barriers to deprescribing than medical students [11]. While 97.4% of respondents believed that pharmacists played a role in deprescribing, the responses also suggested that more training is required due to a lack of confidence in deprescribing [11]. Furthermore, a cross-sectional study of student pharmacists from 132 institutions focused on deprescribing in pharmacy curricula exclusively [12]. The results showed that fewer than half of student pharmacists indicated that deprescribing instruction was part of their non-experiential education or their curriculum prepared them to deprescribe in a clinical setting, while a vast majority reported significant perceived barriers to pharmacist deprescribing [12]. These studies demonstrate the need for more focused, qualitative inquiry on deprescribing education in the pharmacy curriculum. The aim of this study was the qualitative exploration of the views of student pharmacists regarding deprescribing in their pharmacy curricula.

## 2. Methods

### 2.1. Study Design and Theoretical Framework

This study used a focus group discussions (FGD) methodology to facilitate the discussion of student pharmacists’ perceptions on deprescribing [13,14]. The theory of planned behavior (TPB) informed the conceptualization of this study, data collection, and thematic analysis [15]. TPB has been applied in disparate areas of the healthcare field and used to interpret a variety of behaviors [16,17]. For example, in the field of pharmacy, TPB has been utilized as a framework to explain patient or pharmacist behavior in vaccine hesitancy, smoking cessation, and chronic disease states [18,19,20]. While numerous theoretical frameworks or models explain different types of behavior, using TPB is advantageous because it draws upon a person’s intention to participate in a behavior at a definite time and place [15]. TPB has three components: (1) “normative beliefs”; (2) “behavioral beliefs”; and (3) “control beliefs” [15]. These components serve as predictors of a person’s perceived attitude, subjective norms, and control toward a specific action or behavior [15]. Thus, TPB provides vital information about student pharmacists’ salient behavior, such as the act of deprescribing, in addition to normative and control beliefs toward this behavior [15]. TPB informed the conceptualization of the novel FGD guide with the goal of understanding student pharmacists’ views on deprescribing and the development of pharmacy curricula that reflect these needs [15].

### 2.2. Recruitment, Focus Group Structure, and Data Collection

The study recruited student pharmacists from three different colleges/schools of pharmacy located in different geographic regions of the United States. This study was approved by the University of Tennessee Health Science Center (UTHSC) Institutional Review Board (IRB # 21-08234-XM, 2 July 2021), University of New England (IRB # 0821, 2 August 2021) and University of Arizona (IRB# 2021-015-PHPR, 9 August 2021). An email that explained the purpose of the study was sent to all students enrolled in the Pharm.D. curriculum in these colleges, inviting students to participate in an FGD led by two facilitators: an external facilitator (DS), to minimize any biases and influence relative to the participants, and a College of Pharmacy faculty member (AC) who specializes in qualitative research. Both researchers are Ph.D. trained in qualitative data collection and analysis. FGDs were semi-structured and lasted up to two hours in length. Facilitators used an interview guide with specific prompts related to TPB elements to achieve thematic saturation [15]. A sample of the FGD questions are provided in the Appendix A. All FGDs were conducted virtually over three months in the Fall of 2021 [14,21,22].

Student pharmacists provided verbal informed consent to the FGD and received a gift card for their participation in the study. The audio recordings of the FGDs were transcribed verbatim by a professional transcriptionist to minimize bias. Both facilitators took field notes during data collection to note non-verbal expressions and interactions that were used for writing memos during the data analysis process [23].

### 2.3. Data Analysis and Rigor

Thematic analysis, as described by Clarke and Braun, was used to identify, analyze, extract, and report the themes from the data collected [24]. Two researchers (DS and AC) read through the transcripts independently to familiarize themselves with the corpus of data. An inductive coding process was used to identify codes, categories, and themes within and across the FGDs [24]. Two members of the analysis team (DS and AC) reviewed each transcript to develop and refine the codebook using Dedoose^®^ (Manhattan Beach, CA, USA), a qualitative software that facilitates data organization and retrieval. Dedoose^®^ allows all team members to independently code themes while simultaneously sharing codes created by team members, which facilitates rapid code sharing and updating. The coding of each transcript was discussed line by line among team members; consensus and thematic saturation were reached. An audit trail was maintained by using memos that facilitated distinct concepts in their initial codes and categories [23,25]. Themes were then generated based on coded quotations through discussion using the TPB framework.

The rigor of this study was achieved by following the Standards for Reporting Qualitative Research (SPQR) guidelines [26]. For example, the research team met after each FGD and discussed the consistency of the data to ensure saturation was achieved [26]. Furthermore, the team discussed the codes and wrote memos to ensure the inductive codes were not influenced by personal biases [26].

## 3. Results

Three colleges, totaling 1366 student pharmacists across different geographic regions of the U.S., were invited to participate in this study (UTHSC, N = 682; UNE, N = 158, University of Arizona, N = 526). A total of 26 student pharmacists participated in four FGDs. Of the 26 participants, 14 self-identified as male and two declined to state their gender identity. The mean age was 24 years old, with a range from 21 to 37 years old. Out of the 26 participants, 16 were enrolled in the fourth year, seven were enrolled in the third year, and three were enrolled in the second year.

Thematic analysis revealed two major themes. In the first theme, student pharmacists indicated that they possessed limited information about the deprescribing process, which is best illustrated by the following quote: “I think deprescribing as a whole is a gap!” The second theme centered on student pharmacists’ recommendation to increase emphasis on deprescribing in pharmacy curricula. These themes highlight how student pharmacists believe that deprescribing education is vitally important and how they perceive their knowledge of the process of deprescribing to be limited.

### 3.1. Theme 1. Limited Information about Deprescribing: “I Think Deprescribing as a whole Is a Gap!”

Most student pharmacists revealed that they had limited knowledge of deprescribing, which they attributed to the lack of emphasis on deprescribing in their curriculum. Many student pharmacists referenced the Beers’ list, rotation experiences, or didactic courses when discussing the few encounters that they had had with deprescribing. Student pharmacists in the last two years of their pharmacy education indicated having increased knowledge on deprescribing, referencing geriatric and pharmacotherapy courses. The student pharmacists reached a consensus about the very limited deprescribing education that currently exists in the pharmacy curriculum and their limited knowledge about the process of deprescribing. For example, the following excerpts highlight these findings in their discussion about their perspective on deprescribing:


*“I think deprescribing as a whole is a gap! When I was reading the name of this study, it was talking about deprescribing in your curriculum, and I literally was like, when did we talk about deprescribing in the curriculum? And you mentioned like, what tools do you have to reference? I honestly didn’t even know there were tools we could reference. So I think just as a whole it’s a gap”.*
(ST4, FG2, P3)


*“…100% agree with (ST4)…We don’t have any kind of—yeah, the whole thing is a gap, we don’t have any kind of experience with deprescribing in the actual curriculum”. *
(ST2, FG2, P4)


*“Yeah, kind of what ST4 and ST2 were talking about…I was reading an article about deprescribing, I don’t know anything about this…And like there’s no tools, we couldn’t go back on our notes. But I think just the term itself hasn’t been really used in our curriculum”.*
(ST5, FG2, P5)


*“I know like I had mentioned polypharmacy or Beers Criteria and like geriatric studies, that kind of thing obviously hint to deprescribing, and a lot of it is like plugged into our curriculum, but I think being aware of the term itself and what falls below that and kind of referencing that would be really helpful, or else you’ll end up like us, googling and looking at articles right before this study”.*
(ST3, FG2, P4)

Student pharmacists also discussed that they did not have a common definition of “deprescribing.” Some participants attributed their lack of deprescribing knowledge to the absence of didactic lectures on the topic.


*“…there hasn’t been any class or any lectures during class that focused on deprescribing so far…” *
(ST1, FG3, P2)


*“I haven’t personally had, you know, an in-person experience deprescribing. So…I have no experience with deprescribing, so, yeah”.*
(ST3, FG3, P4)


*“…I’m going to be honest… we didn’t really go too much into it. I believe we had two lectures in our therapeutics class really going over general just deprescribing, when it’s a good option to follow that route, and they also gave us some helpful websites as far as how to go about deprescribing some specific medications, but there really wasn’t a lot of in-depth training that I received from that. Honestly, I don’t think it’s really talked about too much. I’m not sure”. *
(ST4, FG1, P4)

One student pharmacist pointed to various resources they were provided with during their education:


*“…we were given different scales, we were given MedStopper, and there was another application, Polypharmacy Guidance, that can all be used when deprescribing”. *
(ST4, FG4, P4)


*“Several students were curious about how deprescribing is taught in other healthcare curricula and, specifically, about the depth of training received by the medical students. This excerpt emphasizes the perceived sense of importance a student placed on training on deprescribing in the pharmacy curriculum and its integration in the healthcare system and care for patients in the context of supporting pharmacist-provided patient care”. *
(ST3, FG1, P4)


*“…I think it’s very important for them [medical students]—for us to know, as well, and for them to know in their curriculum what kind of training they get. Do I know what kind of training they get? No, but I’m assuming that, if we’re going to pick any profession, definitely the prescribers, the ones who are actually writing the medication, need the most extensive training”. *
(ST3, FG1, P4)


*“I would agree, yes, there is a gap… I’m sure how beneficial it would be to know what the other professions are actually taught as far as deprescribing”. *
(ST4, FG1, P4)

### 3.2. Theme 2. Empowering Student Pharmacists through Emphasizing Deprescribing in the Pharmacy Curriculum

This theme describes student pharmacists’ awareness of certain aspects of deprescribing and their recommendations regarding their curricula. A number of students emphasized having specific rotations during their final year of school to provide hands-on experience that could empower them with the knowledge and experience necessary for practice. Furthermore, several student pharmacists highlighted the importance of preceptors in their growth as future providers.


*“I also think it needs to be something that’s brought up to the preceptors, kind of like what (ST3, FG2, P4) was saying is that, I mean, if the preceptor is pushing you to deprescribe and get people off certain medications they don’t need to be on, and it’s every preceptor… has that as part of like maybe their eval or some sort of that, and I think that will also help influence just the natural culture of deprescribing and bring it more to the forefront…And I think that, if-- I mean, and it starts with us, but it also starts with our preceptors, too, because they’re the ones that push us and mold us into the future pharmacist”. *
(ST1, FG2, P4)


*“…I completely agree with ST2, and I completely agree with what ST1 said. Also speaking to ST1′s point, even if we add a lecture-- so we talk about it like oncology… we’re in neuro and psych…And so, even if you add something in, right now we’re learning about mental disorders, so something like a very-- you know, one lecture saying, for the mental disorders, these are the special consideration drugs for deprescribing. So at least somewhere to start, you know, start that lecture and then maybe we can incorporate that into an elective or a longer course, something just to get it started”. *
(ST4, FG2, P3)

## 4. Discussion

The two emerging themes highlight the importance of including deprescribing in pharmacy curricula: student pharmacists had limited knowledge about deprescribing and they recommended increasing the emphasis on deprescribing in pharmacy curricula. Geriatrics education is one practice area where this topic could be seamlessly integrated. Unfortunately, geriatrics is not a required curricular element for the purpose of accreditation, and thus is not always included in pharmacy curricula. This study highlights an opportunity to improve the integration of deprescribing education into pharmacy curricula, both didactic and experiential, and colleges of pharmacy should evaluate whether, where, and to what extent the incorporation of this topic into their curricula is appropriate.

Our findings demonstrate that student pharmacists were not aware of the concept of deprescribing and that there is limited familiarity with the terminology used in deprescribing. These findings align with a survey conducted on this topic by Poots et al. among pharmacy and medical students in the United Kingdom [27]. Poots et al. reported that only 15% of medical and pharmacy students were familiar with the term “deprescribing”; however, 94% stated that they had learned about stopping medications at some point in their education [27]. These findings suggest that greater emphasis is needed on linking the concept and terminology of deprescribing when it is taught in the curriculum and later in experiential activities.

The emergent themes indicate a greater need for research in and instruction on deprescribing to student pharmacists. Recently, an international collaboration was established to develop evidence-based deprescribing guidelines, with input from ten different countries and over 100 institutions [28]. Although some guidelines have been developed and implemented locally, there is a need for the more widespread implementation of these guidelines as well as the development of further guidelines as required [26]. Such efforts are an important first step in improving healthcare providers’ knowledge and skills around deprescribing and developing plans to improve deprescribing education for the future [28]. Such evidence-based guidelines and approaches will facilitate the broadening of deprescribing in practice and subsequently increase student exposure.

Despite the clear need for deprescribing in clinical practice, resources for and research on effective deprescribing are limited. That said, organizations such as the Canadian Deprescribing Network (CADEN) offer several deprescribing guidelines pertaining to anti-hyperglycemics, proton pump inhibitors, benzodiazepine and benzodiazepine-receptor agonists, antipsychotics, and cholinesterase inhibitors and memantine [29]. CADEN also offers “evidence-based pharmaceutical opinion” forms which can be completed by community pharmacists to communicate to prescribers the risks of continuing medications such as sedative-hypnotics or non-steroidal anti-inflammatory drugs for their specific patients [29]. The New South Wales Therapeutic Advisory Group Inc. and Primary Health Tasmania have several deprescribing guides which can be implemented in practice; though, unlike the CADEN resource, they lack evidence-based guidelines to support recommendations [29,30].

Several deprescribing initiatives have been described in the literature, including OPTIMIZE (Optional Medication Management In Alzheimer’s Disease and Dementia) [31], IMPROVE (Initiative to Minimize Pharmaceutical Risk in Older Veterans) [32], and VIONE (Vital, Important, Optional, Not indicated, Every medication has a reason) [33]. These programs offer transparent methods for recreating deprescribing initiatives in most clinical settings.

Bruyère Research Institute offers a free online module on polypharmacy and deprescribing which is geared towards pharmacists and student pharmacists [34]. This module can be offered asynchronously in the pharmacy curriculum and provides a certificate [34]. Most schools of pharmacy provide an introduction to the Beers Criteria^®^ or the STOPP/START criteria, which are explicit tools used to help pharmacists identify potentially inappropriate medications and candidates for deprescribing [35]. However, the Bruyère Research Institute offers an approach to apply these materials to a patient case and guidance on more effectively applying deprescribing techniques [34]. Finally, Medstopper, an online tool used to prioritize medications to discontinue in frail patients, is a free tool that can be used by anyone with internet access [36]. The aforementioned resources are highly accessible and offer a simple way for student pharmacists to engage with deprescribing on a more meaningful level. However, further research is needed to provide additional tools, communication frameworks, and deprescribing guidelines which can be incorporated in pharmacists’ practice, regardless of setting.

Furthermore, in an effort to address the gap in deprescribing knowledge, Pruskowsi et al. described the outcomes of a required, longitudinal residency deprescribing experience for second year postgraduate pharmacy residents at their institution [37]. These residents performed a comprehensive medication review (CMR) with eligible patients in a nursing home, conducted medication-related goals of care conversations with the patient and their caregivers, and made recommendations for deprescribing medications along with monitoring plans [27]. Their report concluded that this focused deprescribing initiative provided opportunities to pharmacy residents to improve their empathy, critical thinking, and communication skills around deprescribing [37]. Colleges of pharmacy could look to this longitudinal initiative as guidance on how to incorporate similar training into their curriculum, with the addition of pharmacist supervision to ensure students have the opportunity to hone their deprescribing skills.

Another approach could include developing a one-day interprofessional workshop that seeks to improve deprescribing knowledge and skills among physicians, physician assistants, nurse practitioners, pharmacists, and clinic staff, as described by Zimmerman et al. [38]. After their workshop, participants indicated an intention to change their practice, teaching, research, or administrative responsibilities related to deprescribing [38]. Perhaps a workshop such as this could help improve the knowledge and skills of those instructing student pharmacists with the intent of improving the teaching of deprescribing to these healthcare professional students.

In another example, Sun et al. developed a competency framework to incorporate deprescribing into the nursing curriculum in Canada [39]. This framework consisted of three major facilitating factors: effective education and training in deprescribing, the need for continuing education and professional development in medication optimization, and the benefits of multi-disciplinary involvement in medication management [39]. Although this was developed for nursing students rather than pharmacy students, a first step may be to develop a competency framework to incorporate deprescribing into pharmacy education, with our findings serving as a foundation for the project [39]. For instance, our findings indicated there was a gap in student pharmacists’ education about deprescribing, and there was interest in understanding the education of other healthcare professionals about deprescribing. Using the concept of the Sun et al.’s study, further research may be warranted to adapt or develop a framework to incorporate deprescribing into the pharmacy curriculum in the US [39].

## 5. Strength, Limitation, and Future Studies

This study was comprised of a heterogeneous sample size of student pharmacists from different geographic areas (South, Southwest, and Eastern parts of the U.S.) that facilitated a broad view on how deprescribing is perceived by student pharmacists. Although the sample size was heterogeneous, the student pharmacists were predominantly third- and fourth-year student pharmacists. Further studies may consider recruiting first- and second-year student pharmacists to better understand how to address their needs.

The qualitative study design and the use of TPB facilitated a novel study design to capture student pharmacist perceptions from a heterogeneous sample. Furthermore, the online platform allowed student pharmacists from different geographic areas to participate and share their views about deprescribing. However, the recruitment and sample characteristics may limit the generalizability of the findings to other U.S. student pharmacists. The findings from the current study highlight the need for more longitudinal studies to explore deprescribing in pharmacy and the curricula of other healthcare programs.

## 6. Conclusions

This study highlights an opportunity to improve the integration of deprescribing education into pharmacy curricula, with two emerging themes: (1) student pharmacists had limited knowledge about deprescribing, and (2) they recommended increasing the emphasis on deprescribing in pharmacy curricula. With the high rate of polypharmacy and the utilization of potentially inappropriate medications, student pharmacists will be expected to deprescribe in clinical settings during experiential education and post-graduation, regardless of the career path they pursue. Geriatrics care education is one practice area where this topic could be seamlessly integrated. Colleges of pharmacy should evaluate whether, where, and to what extent the incorporation of deprescribing into their curricula is appropriate.

## Data Availability

Not applicable.

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
