# Peer review of "“I Think Deprescribing as a Whole Is a Gap!”: A Qualitative Study of Student Pharmacist Perceptions about Deprescribing"

_geriatrics, 2022, doi:10.3390/geriatrics7030060_

Round 1
Reviewer 1 Report
Thus is a much needed research.
Sample
Please state the total number of students approached to participate. How many Pharm D. students are there is the US and what is the generalisability of your findings?
Discussion
Please provide much more critical discussion of strategies, tools and results for deprescribing by pharmacists. Readers need to know the strength of evidence for solutions for this difficult problem.
Author Response
Please state the total number of students approached to participate. How many Pharm D. students are there is the US and what is the generalisability of your findings?
Response: Thank you for this clarification. We sent emails to the three pharmacy schools that were sampled. UTHSC, N= 682; UNE N= 158 , Arizona N=526
However, this is a qualitative study where generalizability cannot be obtained in the similar way to quantitative studies.
The followed the qualitative guidelines regarding the sample size and data generalization.
1. Morse JM. Determining sample size. Sage Publications Sage CA: Thousand Oaks, CA; 2000.
- Morse J. Strategies for sampling in: Morse j. M, editors. Qualitative nursing research: A contemporary dialogue. London: Sage Publications; 1991.
Discussion
Please provide much more critical discussion of strategies, tools and results for deprescribing by pharmacists. Readers need to know the strength of evidence for solutions for this difficult problem.
We agree the audience will likely want to learn about more resources available on this topic. Unfortunately deprescribing is limited in terms of guidelines and tools; however, we have added to the discussion section include resources that focus on deprescribing guidelines or guidance (Deprescribing.org via CADEN, The New South Wales Therapeutic Advisory Group, and Primary Health Tasmania. We have also included examples of deprescribing initiatives such as VIONNE, IMPROVE, and Optimize. Finally, we included information regarding free tools that can be used in the curriculum to help students engage more actively with deprescribing, including the MedStopper tool and the Polypharmacy and Deprescribing certification program offered through the Bruyère Research Institute
Reviewer 2 Report
Thank you for the opportunity to review this manuscript on perceptions of de-prescribing for student pharmacists. De-prescribing is an important topic in geriatric medicine, has significant implications on mortality and costs for patients, and is very pertinent to this journal.
Overview: this manuscript details the study which was a qualitative thematic analysis of focus group discussions of 26 student pharmacists from across the United States and their perceptions on de-prescribing. Two themes emerged: 1. Student pharmacists possess limited information about the de-prescribing process and 2. Student pharmacists recommended increasing emphasis on de-prescribing and pharmacy curricula.
Research question/hypothesis: this appears to be clear and is focused on understanding the perceptions of pharmacy students on the de-prescribing process and trying to get themes from their focus group discussions.
Research design: This is a qualitative study using focus group discussions and thematic analysis of the transcripts of those discussions to understand major themes for pharmacy students on the topic of the de-prescribing. This is therefore descriptive data only. I would suggest providing the questions used by the focus group facilitators or interviewers as described in line 108-109 “facilitators used interview guide with specific prompts”. If editors agree this could be in an appendix to this article.
Data analysis and statistics: the authors follow the standards for reporting qualitative research (SRQR). They describe their approach adequately in their method section. In their data analysis, the process which they used to discover themes was using the theory of planned behavior framework with a well described coding schema which categorized the various quotes from the transcript into themes. There is an ongoing process to ensure this is not influenced by personal biases.
Results: Results are presented as the two major themes (lack of knowledge of the prescribing and a need to address this in their curricula) and very useful quotes to support these themes.
Discussion and conclusion: The discussion and conclusion section are clearly stated and their conclusions follow from the results. They give suggestions on addressing the deficit of de-prescribing knowledge for pharmacy students. Specifically, they call for more integrated geriatrics education with pharmacy students, student participations in the prescribing initiatives, and workshops. These findings are appropriate in the context of the relevant literature. Limitations of the study are appropriately addressed.
Author Response
Research design: This is a qualitative study using focus group discussions and thematic analysis of the transcripts of those discussions to understand major themes for pharmacy students on the topic of the de-prescribing. This is therefore descriptive data only. I would suggest providing the questions used by the focus group facilitators or interviewers as described in line 108-109 “facilitators used interview guide with specific prompts”. If editors agree this could be in an appendix to this article.
Response: Thank you for this suggestion. We amended the text to include the FG guide as an appendix.
Data analysis and statistics: the authors follow the standards for reporting qualitative research (SRQR). They describe their approach adequately in their method section. In their data analysis, the process which they used to discover themes was using the theory of planned behavior framework with a well described coding schema which categorized the various quotes from the transcript into themes. There is an ongoing process to ensure this is not influenced by personal biases.
Response: Thank you for this comment. Indeed, the team ensured the analysis process was not influenced by personal biases.
Results: Results are presented as the two major themes (lack of knowledge of the prescribing and a need to address this in their curricula) and very useful quotes to support these themes.
Discussion and conclusion: The discussion and conclusion section are clearly stated and their conclusions follow from the results. They give suggestions on addressing the deficit of de-prescribing knowledge for pharmacy students. Specifically, they call for more integrated geriatrics education with pharmacy students, student participations in the prescribing initiatives, and workshops. These findings are appropriate in the context of the relevant literature. Limitations of the study are appropriately addressed.
Response: Thank you for your suggestions. The results will help the pharmacy educators to develop appropriate workshops to integrate deprescribing initiatives.
Round 2
Reviewer 1 Report
Thanks to the authors for the total number of pharmacy students at the three universities. [the three pharmacy schools that were sampled. UTHSC, N= 682; UNE N= 158 , Arizona N=526]Include these numbers in the Abstract, Text and Conclusions with a disclaimer that with such small numbers and small %s of eligibles no generalisability is possible.
Author Response
Thanks to the authors for the total number of pharmacy students at the three universities. [the three pharmacy schools that were sampled. UTHSC, N= 682; UNE N= 158 , Arizona N=526]Include these numbers in the Abstract, Text and Conclusions with a disclaimer that with such small numbers and small %s of eligibles no generalisability is possible.
Response: Thank you for your suggestions. We amended the text.